# *Leap-of-Thought*: Accelerating Transformers via Dynamic Token Routing

**Yeachan Kim[1], Junho Kim[1], Jun-Hyung Park[2], Mingyu Lee[1], SangKeun Lee[1,3]**
[1]Department of Artificial Intelligence, Korea University, Seoul, South Korea
[2]BK21 FOUR R&E Center for Artificial Intelligence, Korea University, Seoul, South Korea
[3]Department of Computer Science and Engineering, Korea University, Seoul, South Korea
{yeachan,monocrat,irish07,decon9201,yalphy}@korea.ac.kr

## Abstract

Computational inefficiency in transformers has been a long-standing challenge, hindering the deployment in resource-constrained or real-time applications. One promising approach to mitigate this limitation is to progressively remove less significant tokens, given that the sequence length strongly contributes to the inefficiency. However, this approach entails a potential risk of losing crucial information due to the irrevocable nature of token removal. In this paper, we introduce **L**eap-**o**f-**T**hought (LoT), a novel token reduction approach that dynamically routes tokens within layers. Unlike previous work that irrevocably discards tokens, LoT enables tokens to 'leap' across layers. This ensures that all tokens remain accessible in subsequent layers while reducing the number of tokens processed within layers. We achieve this by pairing the transformer with dynamic token routers, which learn to selectively process tokens essential for the task. Evaluation results clearly show that LoT achieves substantial improvement on computational efficiency. Specifically, LoT attains up to $25\times$ faster inference time without a significant loss in accuracy[1].

## 1 Introduction

The advent of Transformer (Vaswani et al., 2017) has spurred a paradigm shift, most notably in natural language processing (Brown et al., 2020; Chowdhery et al., 2022), but also extending to computer vision (Dosovitskiy et al., 2021; Liu et al., 2021). However, the impressive capabilities of the transformer typically come with non-trivial computational costs, which scale quadratically with the length of the input sequence. This computational burden poses a significant challenge when deploying the transformer-based models in resource-constrained or real-time systems (Sun et al., 2020).

One typical approach to tackle this challenge is to reduce the number of tokens processed within

transformer layers (Goyal et al., 2020; Ye et al., 2021; Guan et al., 2022). The rationales behind this approach are two-fold: (i) not all tokens are equally significant to the task (Dai et al., 2020), and (ii) all token representations gradually become similar over layers (Abnar and Zuidema, 2020; Phang et al., 2021). Based on these rationales, previous studies progressively removes the less significant or redundant tokens (Figure 1a), selected either randomly (Hou et al., 2022) or based on the attention scores (Wang et al., 2019). However, the permanent token removal in the previous work could entail a risk of discarding crucial information in pursuit of efficiency, which can potentially degrade performance by hindering the fine-grained understanding of the input. Moreover, since the token reduction space with the permanent removal is proportionally constrained with the number of remaining tokens, it is sub-optimal to explore the diverse reduction strategies that potentially offer a greater efficiency. These limitations suggest that there is still room for further improvement.

In this paper, we propose **L**eap-**o**f-**T**hought (LoT)[2], a novel token reduction approach that enables the dynamic routing of tokens across layers. In contrast to permanent removal strategies, LoT allows the tokens to 'leap' over each layer, thereby retaining access to all original tokens in the subsequent layers while reducing the number of tokens processed within each layer of the transformer (Figure 1c). We achieve this by coupling the transformer layers with dynamic token routers, which learns to decide whether the given token should be processed at the current layer or leaped forward to the next layer. Moreover, in order to steer the token router towards making informed and efficient decisions, we introduce a gradient-guided training

---

[2]The name comes from the LoT behavior where each token (corresponding meaning or thought) leaps over layers. LoT is not related to the Chain-of-Thought (Wei et al., 2022), which introduces the concept of continuous prompting.

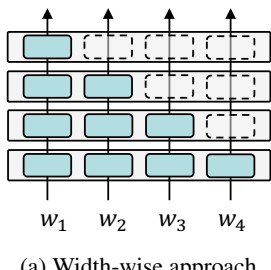  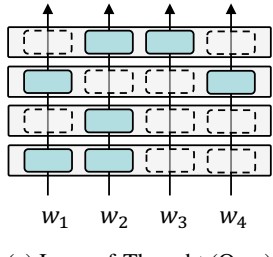

| (a) Width-wise approach | (b) Depth-wise approach | (c) Leap-of-Thought (Ours) |

Figure 1: Reduction strategies of width-wise (token reduction), depth-wise (layer reduction) and Leap-of-Thought (ours) to achieve computational efficiency. The tokens with the dashed lines indicate the unused tokens in each layer.

that informs each router of which tokens do significantly contribute. Consequently, the router learns to identify which tokens are crucial for the task and where these tokens should be processed within layers in order to achieve greater efficiency.

LoT offers several advantages compared to the permanent removal. Primarily, LoT has the potential to mitigate the risk of losing crucial information related to the task, given that the decisions for each token are recoverable in subsequent layers. In addition, LoT provides a higher degree of freedom in token reduction, thereby facilitating the exploration of a diverse search space for greater efficiency, which is similarly observed in network compression (Mao et al., 2017; Park et al., 2023).

To substantiate the efficacy of LoT, we perform evaluations across extensive experiments. Comprehensive results demonstrate that the model employing LoT reveals substantial speedup gains without a significant loss in task accuracy. Additionally, through the analysis of LoT, we provide justification for the efficacy of the dynamic token routing mechanism and illustrate how LoT achieves greater efficiency. In summary, the contributions of the paper include the followings:

- We introduce Leap-of-Thought, a novel token reduction approach that enables dynamic token routing within the transformer, which reduces the processed tokens within each layer while preserving crucial information.

- We propose a gradient-guided training to steer the dynamic token router towards making more informed decisions about whether the tokens should be processed or leaped over.

- We demonstrate the efficacy of LoT through extensive experiments and analysis on various benchmarks, establishing LoT as a promising approach for the token reduction.

## 2 Related Work

In this section, we mainly review the methods that adaptively control the computation in pre-trained language models. Recent approaches can be classified into two categories: width-wise and depth-wise approaches. The former focuses on reducing the number of tokens processed by transformers, while the latter aims to decrease the number of computational layers. Figure 1 illustrates the distinct behaviors of these approaches, including LoT.

### 2.1 Width-wise Reduction on Transformer

Given that the computational costs of the transformer are heavily influenced by the length of the input sequence (Tay et al., 2023), recent research has endeavored to minimize the sequence length by progressively removing less significant tokens from the input (Figure 1a). For instance, PoWER-BERT (Goyal et al., 2020) have initially explored the removal of tokens that receive the least attention from other words, operating on the premise that tokens with less attention are less significant. However, several studies on the transformer interpretability have shown that the attention scores might not be reliable indicators of the actual token contribution (Jain and Wallace, 2019; Abnar and Zuidema, 2020; Meister et al., 2021). As such, TR-BERT (Ye et al., 2021) and Transkimmer (Guan et al., 2022) have suggested token removal strategies that can be learned during training, by using reinforcement learning and reparameterization tricks, respectively. Subsequently, AdapLeR (Modarressi et al., 2022) have proposed a saliency-based strategy that eliminates tokens by estimating the saliency of tokens via the gradients of the input embeddings with respect to the predictions.

While these methods have demonstrated efficiency in downstream tasks, they irrevocably dis-

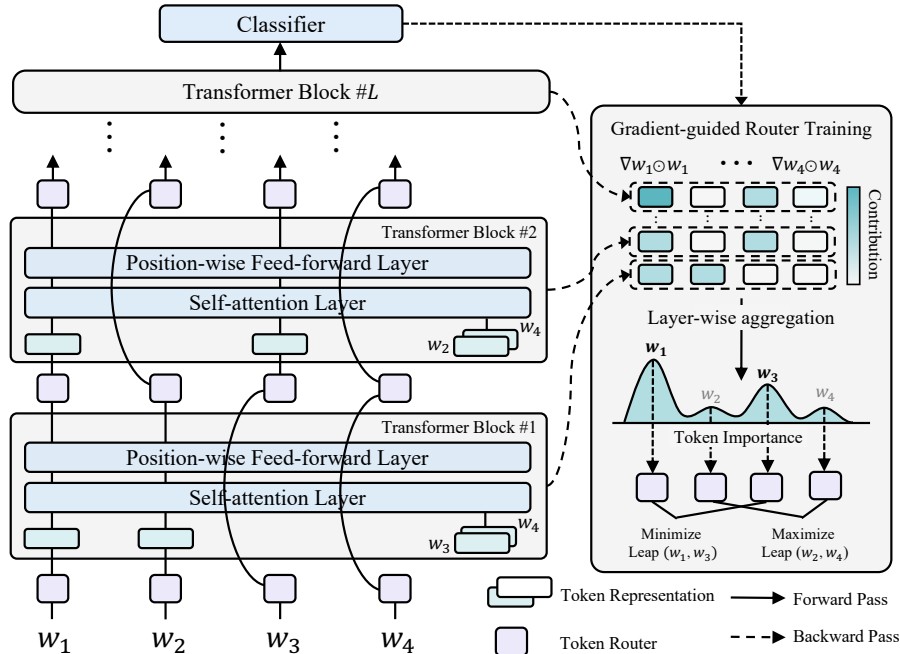

Figure 2: Overview of Leap-of-Thought (LoT). Starting from the embedding layer, the token routers are located between every transformer layers. For the words that are decided to be leaped forward to the next layer, their representations are merged into one pseudo token to provide the minimal information of the unused tokens.

card input tokens, which might lead to a potential loss of crucial information. Moreover, the search space for token removal is proportionally constrained by the number of remaining tokens, thereby restricting flexibility in optimizing reduction strategies. In contrast, since LoT allows the model to revisit all tokens, the crucial information can be better preserved within the transformer. Besides, the ability to revisit tokens endows LoT with a higher degree of flexibility in exploring diverse reduction space that potentially offers greater efficiency.

## 2.2 Depth-wise Reduction on Transformer

The principle behind depth-wise approach is to allocate minimal layer computations to easy samples while dedicating more layer computations to difficult samples (Figure 1b). The distinction between different works lies in the criteria for difficulty. PABEE (Zhou et al., 2020) has proposed a patience-based exit strategy that halts the forward-pass at an intermediate layer only when the pre-defined number of subsequent layers yield the same predictions. Similarly, DeeBERT (Xin et al., 2020) and FastBERT (Liu et al., 2020) have employed the predictive entropy to replace the patience, and PCEE-BERT (Zhang et al., 2022) has combined both patience and confidence for the exit criteria.

Instead of implementing an exit strategy, Layer-Drop (Fan et al., 2020) has demonstrated its efficiency by randomly dropping the layers, and BERT-of-Theseus (Xu et al., 2020) has learned to replace the subsequent two layers with a single layer.

While these works allow adaptive computation on different inputs to achieve efficiency, the level of granularity in the depth-wise approach is constrained by the number of layers. This could result in the sub-optimal efficiency and difficulty in assigning fine-grained computations to a diverse set of samples.

## 3 *Leap-of-Thought*: Dynamic Token Routing for Accelerating Transformer

In this section, we elaborate on Leap of Thought (LoT), which dynamically routes tokens across layers to improve computational efficiency. To this end, we introduce a dynamic token router in learning to decide which token should be processed in the current layer or leaped forward to the subsequent layer (Section 3.1). To ensure that the token router makes well-informed decisions, each token router is trained by a gradient-based token importance (Section 3.2). The overall process of LoT is illustrated in Figure 2.

## 3.1 Learning to Leap Transformer Layer

In order to enable tokens to leap across transformer layers, we introduce a dynamic token routing mechanism that adaptively selects tokens for utilizing in the current layer, while pushing the unused tokens forward to subsequent layers for potential use.

**Dynamic Token Router.**   To initiate the routing mechanism, we start by the definition of a dynamic token router, a lightweight module located between every transformer layers. Each router takes token representations as the input (i.e., embedding or outputs from the previous layer) and learns to produce a binary decision for each token: "1" denotes that it is processed at the current layer, and "0" denotes that it leaps to the next layer. The dynamic token router is formulated as follows:

$$u(w) = \sigma_2(W_2\sigma_1(W_1(LN(w)) + b_1) + b_2) \quad (1)$$

where $w$ is a token representation, $W$ and $b$ denote the weights and biases for linear transformation, respectively, $\sigma_1$ and $\sigma_2$ indicate the GeLU activation and softmax function, respectively, and $LN(\cdot)$ denotes the layer normalization (Ba et al., 2016)[3]. We then derive the routing decision based on the prediction of the router.

$$\mathcal{R}(w) = \begin{cases} 1 & \text{if } u_{\text{process}}(w) > u_{\text{leap}}(w) \\ 0 & \text{otherwise,} \end{cases} \quad (2)$$

where the subscript of $u(w)$ represents the probability for each actions (i.e., process or leap the layer).

**Routing Tokens.**   Once the token router is established, the routing decision is applied to all tokens before they are fed into the transformer computation. Formally, let the token representations in the $l$-th layer be denoted as $w_0^{(l)}, w_1^{(l)}, ..., w_{n-1}^{(l)}$, where $n$ is the length of an input. The routing decision is made for each token[4] by applying the following gating function.

$$w_i^{(l)} = \mathcal{R}^{(l)}(w_i^{(l)} + c^{(l)}) \odot w_i^{(l)}, \quad (3)$$

where $\mathcal{R}^{(l)}(\cdot)$ is the routing function on the $l$-th layer, $\odot$ indicates the Hadamard product, and $c^{(l)}$ is the context vector used to make the routing decision by considering the current context information.

---

[3]We empirically observed that applying layer normalization makes the training of LoT more stable.

[4]Note that the [CLS] token is not forwarded to the router to perform the classification task correctly.

Notably, we employ the [CLS] token (i.e., $w_0^{(l)}$) as the context vector, given that it serves as a contextual memory, being retained throughout all layers.

However, training the router in an end-to-end manner is non-trivial due to the non-differentiable nature of the routing function. To circumvent this, we utilize the Gumbel-softmax reparameterization (Jang et al., 2017) to approximate the discrete decisions during training. Specifically, we introduce a continuous relaxation for the discrete decision-making process. During the forward pass, we sample from a Gumbel distribution and apply the softmax function to approximate the discrete decisions

$$u(w) = \text{softmax}\left((\log(u(w)) + g)/\tau\right), \quad (4)$$

where $g$ is a sample from a Gumbel distribution, and $\tau$ is the temperature parameter controlling the smoothness of the approximation. During the backward pass, we replace the gradient of the non-differentiable function with that of the Gumbel-softmax using straight-through-estimator (Bengio et al., 2013). This allows gradients to flow through the router, enabling end-to-end optimization of the entire model.

**Token Merging.**   While the routing ability allows the model to preserve crucial information, maintaining the minimal information of unused tokens can be beneficial. We thus introduce token merging mechanism. Formally, the merged token is constructed as follows:

$$w_{\text{merge}}^{(l)} = \frac{1}{m} \sum_{i=1}^{n-1} \mathbb{1}[\mathcal{R}^{(l)}(w_i^{(l)} + c^{(l)}) = 0]w_i^{(l)}, \quad (5)$$

where $\mathbb{1}[x]$ is the indicator function that returns one if the statement $x$ is true; otherwise zero, and $m$ is the number of tokens to be leaped. The merged token is appended to the input and only utilized in the self-attention layer. In the next layer, the token is replaced with a new merged token based on the new routing results (i.e., $\mathcal{R}^{(l+1)}$).

## 3.2 Gradient-guided Router Training

To steer the token router towards making informed decisions, we also introduce a gradient-guided router training, which directly provides the supervision of the significant tokens to the routers.

**Guidance Derivation.**   As a guidance for the router, the gradients of the token representations are leveraged, given that the gradient information

can encode the sensitivity of the output to the input tokens, providing insight into which tokens are being more influential for prediction (Jain and Wallace, 2019; Abnar and Zuidema, 2020). Inspired by Grad-CAM (gradient-weighted class activation map) (Selvaraju et al., 2017) which uses gradient information flowing into the final layer of convolutional neural networks, we derive class activation tokens on each each layer. Formally, let $y$ be the prediction for the ground-truth, the class activation tokens (CAT) can be derived as follows:

$$\text{CAT}_i^{(l)} = \frac{\partial y}{\partial w_i^{(l)}} \odot w_i^{(l)}, \qquad (6)$$

Based on the gradient-weighted token representations, we derive the importance by the magnitude of each CAT. Specifically, we aggregate the token importance from all layers since it can provide a better identification for the important tokens (Qiang et al., 2022) (detailed in Section 5.1):

$$\text{CAT}_i = \sum_{l=0}^{L-1} \|\text{CAT}_i^{(l)}\|_2 \qquad (7)$$

Lastly, we need to identify which range of token importance should be considered as significant. To this end, we simply select the tokens whose cumulative sum of their sorted and normalized importance scores falls below a pre-defined threshold $p$, similar to the candidate set of nucleus sampling (Holtzman et al., 2020).

**Training Objective.** The dynamic token routers are trained to process only the significant tokens which are selected from the above procedure. Let $\hat{w}_i$ be the selection decision for the $i$-th token given the selected tokens with a value of one otherwise zero, the objective for the router is formulated as follows:

$$\mathcal{L}_{\text{router}}^{(l)} = -\sum_{i=1}^{n-1} \hat{w}_i \cdot \log(u_{\text{process}}(w_i^{(l)}))$$
$$+ (1 - \hat{w}_i) \cdot \log(u_{\text{leap}}(w_i^{(l)})), \quad (8)$$

The overall objective function for the downstream task can be formulated as follows:

$$\mathcal{L} = \mathcal{L}_{\text{task}} + \frac{\lambda}{L} \sum_{l=0}^{L-1} \mathcal{L}_{\text{router}}^{(l)}. \qquad (9)$$

where $\mathcal{L}_{\text{task}}$ is the task-specific loss function (e.g., cross entropy for the classification), and a harmony coefficient $\lambda$ to balance the two loss terms.

# 4 Experiment

In this section, we evaluate the proposed method on a series of downstream tasks. We specifically demonstrate that introducing the leap action results in a more favorable computational efficiency compared to the prior methods.

## 4.1 Experimental Setup

### 4.1.1 Datasets

We perform diverse tasks to verify the general applicability. These tasks involve scenarios where the model needs to comprehend a single sequence, as well as cases that requires understanding the semantic relationship between multiple sequences. For the single input tasks, we use SST-2 (Socher et al., 2013) and IMDB (Maas et al., 2011) for sentiment analysis, AG's news (Zhang et al., 2015) for topic classification, DBpedia (Lehmann et al., 2015) for ontology classification, and HateXplain (Mathew et al., 2021) for hate speech detection. For the multiple input tasks, we perform paraphrasing tasks on MRPC (Dolan and Brockett, 2005) and natural language inference tasks on MNLI (Williams et al., 2018) and QNLI (Rajpurkar et al., 2016).

### 4.1.2 Baselines

Following the prior work, we use the pre-trained BERT$_{\text{base}}$ (Devlin et al., 2019) as a backbone network[5]. We then compare with six baselines including the backbone model: PoWER-BERT (Goyal et al., 2020) which utilizes the attention maps to eliminate the tokens; TR-BERT (Ye et al., 2021) that adopts reinforcement learning to learn a removal strategy; AdapLeR (Modarressi et al., 2022) that utilizes the saliency maps of the input words to remove tokens. Additionally, we also compare LoT with different direction of reduction approaches. We compare PCEE-BERT (Zhang et al., 2022), which adaptively exits from the transformer by considering both the confidence and patience, and DistilBERT (Sanh et al., 2019), which is the resultant model from knowledge distillation.

### 4.1.3 Evaluation Metrics

Following the recent prior work (Modarressi et al., 2022), we evaluate each method using both the task accuracy and the number of floating-operations (FLOPs). Given that FLOPs are independent of

---

[5]In Appendix, we included the experiments on different architectures (TinyBERT, BERT$_{\text{large}}$)

Table 1: Evaluation results of test accuracy (%) and speedup ratio on the single input tasks. The speedup ratio (denoted as **Speed**) is computed by comparing the FLOPs of each baseline with the backbone. The best and second best results are highlighted in **boldface** and underlined, respectively.

| Method | SST-2 Acc. | SST-2 Speed | IMDB Acc. | IMDB Speed | HateXplain Acc. | HateXplain Speed | AG's news Acc. | AG's news Speed | DBpedia Acc. | DBpedia Speed |
|---|---|---|---|---|---|---|---|---|---|---|
| Baseline | 92.7 | 1.00× | 93.8 | 1.00× | 68.3 | 1.00× | 94.4 | 1.00× | 99.3 | 1.00× |
| DistilBERT (Sanh et al., 2019) | 92.2 | 2.00× | 92.4 | 2.00× | 68.4 | 2.00× | **94.2** | 2.00× | 99.0 | 2.00× |
| PCEE-BERT (Zhang et al., 2022) | 91.9 | 1.56× | 92.3 | 2.63× | 67.9 | 3.09× | 93.4 | 5.54× | 99.0 | 5.80× |
| PoWER-BERT (Goyal et al., 2020) | 92.1 | 1.18× | 92.3 | 1.70× | 66.9 | 2.69× | 92.1 | 12.50× | 98.1 | 14.80× |
| TR-BERT (Ye et al., 2021) | 92.1 | 1.46× | **93.2** | 2.90× | 67.9 | 2.23× | 93.2 | 10.20× | 98.9 | 10.01× |
| AdapLeR (Modarressi et al., 2022) | 92.3 | 1.49× | 91.7 | 3.21× | 68.6 | 4.73× | 92.5 | 17.10× | 98.9 | **22.23×** |
| LoT (ours) | **92.9** | 2.30× | 92.4 | **3.84×** | **68.8** | **5.21×** | 92.4 | **25.10×** | **99.1** | 19.76× |

Table 2: Evaluation results of test accuracy (%) on multiple input tasks. The best and second best results are highlighted in **boldface** and underlined, respectively.

| Method | MRPC F1. | MRPC Speed | MNLI Acc. | MNLI Speed | QNLI Acc. | QNLI Speed |
|---|---|---|---|---|---|---|
| Baseline | 87.5 | 1.00× | 84.2 | 1.00× | 90.3 | 1.00× |
| DistilBERT | 87.7 | 2.00× | 82.0 | 2.00× | 87.9 | 2.00× |
| PCEE-BERT | 87.2 | 1.34× | 82.5 | 1.10× | **90.4** | 1.31× |
| PoWER-BERT | 88.0 | 1.07× | 82.9 | 1.10× | 89.7 | 1.23× |
| TR-BERT | 81.9 | 1.16× | **84.8** | 1.00× | 89.0 | 1.09× |
| AdapLeR | 87.6 | 1.27× | 82.9 | 1.42× | 89.3 | 1.47× |
| LoT (ours) | **88.4** | **3.29×** | 83.1 | **2.53×** | 90.2 | **2.74×** |

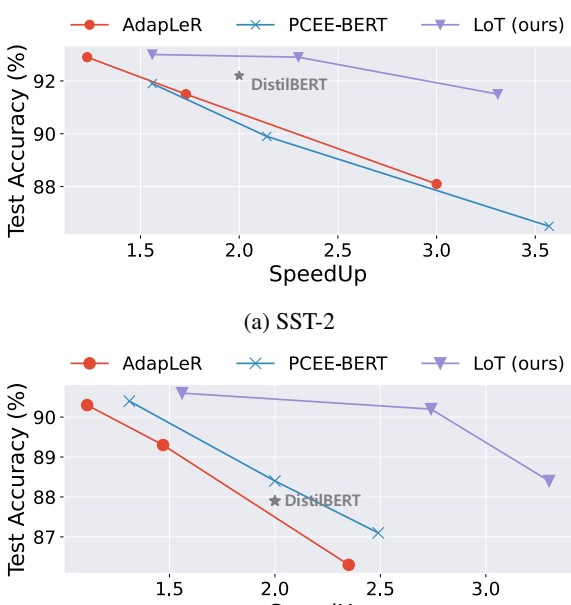

(a) SST-2

(b) QNLI

Figure 3: Trade-off curve between task accuracy and speedup on SST and QNLI datasets.

hardware, this allows us to evaluate the acceleration of the model without taking into account the operating environment[6]. Furthermore, following the recent practice (Guan et al., 2022; Modarressi et al., 2022), we compute FLOPs for a single inference, which enables us to evaluate per-example inference and avoid pseudo speed-up resulting from the elimination of padding tokens. In the experiments, we present the relative speed-up compared to that of the $BERT_{base}$.

### 4.1.4 Training Details

We implement the proposed method using PyTorch. For the hyper-parameters associated with LoT (i.e., threshold $p$ in Eq. (8), and $\lambda$ in Eq (9)), we search the best practice parameters on the validation sets. The hyper-parameters are listed in the Appendix.

### 4.2 Main Results

**Singe input tasks.** In Table 1, we first present the evaluation results on the single input tasks along with strong baselines. Notably, LoT achieves substantial speedup gains without compromising the

---

[6]We also calculated wall-clock inference time in Appendix.

accuracy. For example, LoT achieves speedup of 25.10× on the AG's news dataset without a significant loss in task accuracy. Such a speedup with comparable performance to other baselines demonstrates that LoT better preserves the crucial information related to tasks. This result clearly supports the significance of the leap action in the token reduction approach.

**Multiple input tasks.** We also highlight the results on the tasks that involve pairs of distinct sentences in Table 2. Given that these tasks require the model to comprehend the semantic relationships between multiple sentences, removing tokens could lead to the loss of important information needed for

Table 3: Ablation study of LoT on SST-2 and MRPC, and 'w/o' indicates the model without the corresponding component. The token merging is related to Eq. (5), and the layer-wise aggregation of CAT is related to Eq. (7).

| Method | SST-2 | | MRPC | |
| --- | --- | --- | --- | --- |
| | Acc. | SpeedUp | F1. | SpeedUp |
| LoT (ours) | 92.9 | 2.30× | 88.4 | 3.29× |
| w/o token merging | 92.4 | 2.16× | 88.0 | 2.94× |
| w/o layer-wise CAT | 92.3 | 1.96× | 88.6 | 2.46× |

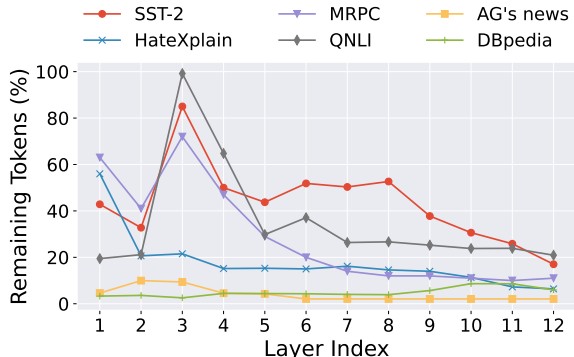

Figure 4: Remaining token distribution across various layers and datasets, excluding [PAD] tokens that may lead to a pseudo speedup.

understanding these relationships. Hence the existing methods, which permanently remove the tokens, reveal the low speedup gains on these datasets. Nevertheless, LoT achieves greater speedup gains with comparable accuracy, which shows the versatility of LoT on diverse tasks. The overall experimental results verify that LoT can be successfully applied into real-world applications that demand both accuracy and efficiency.

**Trade-off.** To confirm the better computational efficiency of LoT, we show the trade-off curves between task accuracy and speedup gains on two representative datasets in Figure 3. This shows that LoT maintains the higher accuracy over a wide range of speedups, clearly demonstrating the better trade-off of LoT compared to other approaches.

## 5 Analysis

In this section, we analyze the behavior of LoT in detail. We specifically focus on how LoT achieves a greater efficiency than other baselines.

### 5.1 Ablation Study

In Table 3, we present an ablation study to dissect the contributions of components in LoT. We

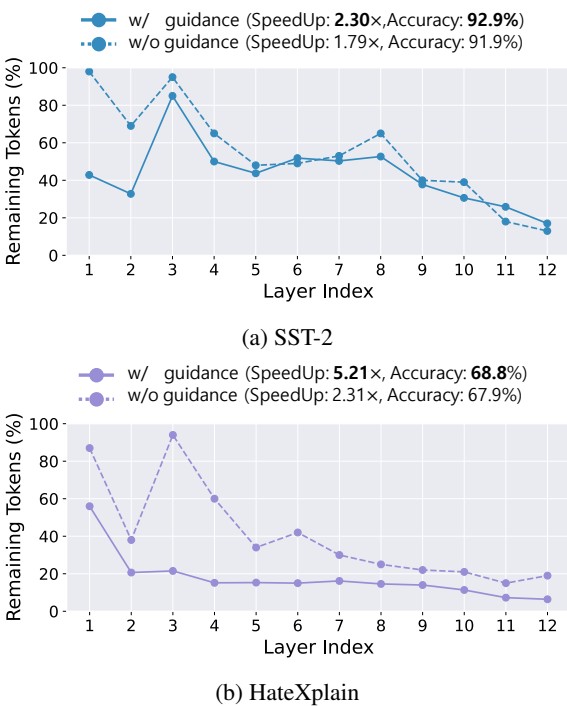

(a) SST-2

(b) HateXplain

Figure 5: Comparison with a LoT variant that learns the leap action in an end-to-end manner.

initially observe that merging tokens enhances performance on both metrics. This indicates that providing abstractive information about tokens to be leaped enables the model to process more reduced tokens in each layer, thereby leading to the improved efficiency. On the other hand, in contrast to GradCAM (Selvaraju et al., 2017), which utilizes gradient information from the input space, LoT aggregates gradient information from all layers. To evaluate the impact of the aggregation, we compare the performance when extracting gradient information solely from the input. The ablation shows that aggregating CAT from all layers substantially improves the computational efficiency, indicating that the aggregation allows the model to better identify tokens that are crucial to the task at hand.

### 5.2 Routing Distribution on Different Layers

We also analyze the routing distribution across different layers for various datasets. Figure 4 shows the ratio of remaining tokens in each layer. Interestingly, we observe distinctive patterns of LoT for the reduction strategy. In the majority of datasets, less than half of the tokens are processed in the first layer. Subsequently, the behavior of LoT varies depending on the complexity of the tasks. For the simpler tasks (i.e., achieved higher speedup gains), the number of the processed tokens tend to

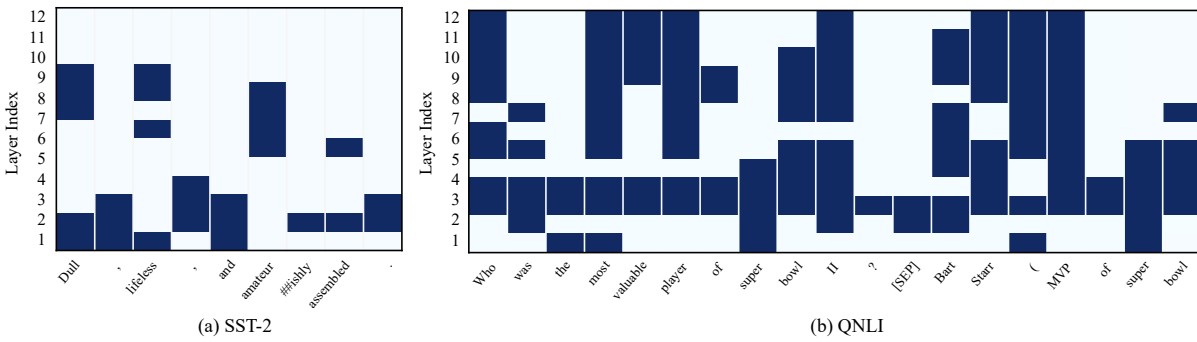

(a) SST-2                    (b) QNLI

Figure 6: Illustration of the token routing on two examples. The darker block in each layer indicates the use of corresponding token in the layer while the lighter block denotes the leap action.

consistently decrease over the subsequent layers. Conversely, for more challenging tasks (e.g., SST, MRPC, QNLI), the model tends to make use of a larger number of tokens in few subsequent layers. These patterns indicate that LoT is capable of adaptively executing optimized reduction strategies for different datasets, underscoring the flexibility and diversified search space of LoT.

## 5.3 Significance of Router Guidance

The token routers in LoT are supervised directly from the aggregated gradient information. To verify the significance of the supervised router training, we compare LoT with an alternative version that learns to decide the leap action without the guidance. To implement this baseline, we replace the guidance loss (i.e., Eq. (8)) with a regularization loss that enforces leap actions for all tokens, which is similarly used in previous work (Ye et al., 2021; Guan et al., 2022; Modarressi et al., 2022). Since this regularization term conflicts with the task objective, the router could learn to retain only the contributing tokens while bypassing the less contributing tokens. Figure 5 shows the comparison result. We observe that the model without guidance can also achieve computational efficiency by learning to leap tokens. However, explicit supervision on the important tokens enables the router to bypass a greater number of tokens, especially in the earlier layers. This phenomenon can be attributed to the low learning capacities of the earlier layers in identifying significant tokens without an explicit guidance. The overall results empirically justify the significance of the gradient-guided router training.

## 5.4 Case Study of Leap-of-Thought

Lastly, we examine the behavior of LoT through case studies. Figure 6 exhibits the routing exam-

ples on two different tasks, SST-2 and QNLI. It is evident that the routing maps are irregular and sparse, which demonstrates the flexibility of LoT to reach greater efficiency. Moreover, the important tokens related to tasks (e.g., *dull*, *lifeless* and *amateur* in SST-2) tend to be utilized in the deeper layers, whereas less contributing tokens are often bypassed. Interestingly, those important tokens are not consistently used in all layers. For example, the sentiment-related words in SST-2 are temporally used in the earlier layers and then reused after several layers. This result highlights LoT's distinctive strategy in optimizing computational efficiency by selectively engaging with task-relevant tokens as needed. We provide the additional case study in Section F of Appendix.

## 6 Conclusion

In this work, we have proposed Leap-of-Thought (LoT), a novel token reduction strategy that enables the dynamic routing of tokens within the transformer layers. Unlike the previous works that permanently remove tokens, LoT learns to decide whether the given token should be processed in the current layer or leaped forward to the next layer. This ensures that all tokens remain accessible in subsequent layers while reducing the number of tokens processed within layers. Through the guidance from the gradient information, each router learns to process only the significant tokens to the task while bypassing the less contributing tokens. The comprehensive evaluations have convincingly supported the superiority of the proposed method by showing that LoT achieves substantial speedup gains over state-of-the-art methods with the comparable task accuracy. The analysis also have strongly supported that introducing the leap action leads to the substantially improved efficiency.

## Limitations

While the proposed method allows transformer-based pre-trained models to achieve greater computational efficiency, there are a few potential limitations.

**- Interpretability** Several existing methods for interpretability, such as layer-wise analysis (Tenney et al., 2019), might not be compatible with our method, given that LoT benefits from the irregularity. As an alternative, we believe that aggregating routing results across layers can serve as a reliable indicator of interpretability to a certain extent. For example, in the case study (Figure 6), while there are irregularity on individual layers, the model tends to frequently use the task-related tokens across layers, such as *dull*, *lifeless*, and *amateur* in the first sample (sentiment analysis), and *who*, *most*, *player* (in the question part of the pair), *Bart*, *Starr*, *MVP* (in the answer part of the pair) in the second example (natural language inference). Such a comprehensive analysis across layers can provide some degree of interpretability for the predictions.

**- Router Overhead** In comparison to the vanilla backbone, LoT employs token routers to perform the dynamic computation, which imposes extra model parameters and computation overhead, similar to other baselines (Ye et al., 2021; Modarressi et al., 2022; Zhang et al., 2022). This is the reason why we have carefully designed the routers as lightweight modules that only account for 2.1% (0.17% for each router) and 2.2% (0.18% for each router) of the memory and computational overhead of the entire model, respectively. To understand the effect of the router capacity, we analyze the trade-off between the computational overhead and total performance in Section D of Appendix. The result shows that such a lightweight router is sufficient to achieve significant speedup gains without compromising task accuracy. Nevertheless, in this paper, we confirm the applicability of such a lightweight router only in the natural language understanding tasks. Therefore, designing routers for natural language generation tasks (e.g., summarization, machine translation) can be a promising avenue for future research.

## Acknowledgment

This work was supported by the Basic Research Program through the National Research Foundation of Korea (NRF) grant funded by the Korea government (MSIT) (2021R1A2C3010430) and Institute of Information & Communications Technology Planning & Evaluation (IITP) grant funded by the Korea government (MSIT) (No.2019-0-00079, Artificial Intelligence Graduate School Program (Korea University)).

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

# Appendix

## A  Dataset statistics

We provide the statistics of the dataset in Table A.

Table 4: Statistics of the datasets used in evaluations.

| | Dataset | Average length | Number of train/test data | Number of classes |
|---|---|---|---|---|
| Single Input | SST-2 | 14 | 70k / 1.8k | 2 |
| | IMDB | 275 | 25k / 25k | 2 |
| | HateXplain | 30 | 15.4k / 1.9k | 3 |
| | AG's news | 53 | 120k / 7.6k | 4 |
| | DBpedia | 64 | 560k / 70k | 14 |
| Multiple Input | MRPC | 53 | 3.6k / 1.7k | 2 |
| | MNLI | 40 | 390K / 9.7k | 2 |
| | QNLI | 50 | 104k / 5.4K | 2 |

## B  Selected Hyper-Parameters

In table 5, we present the selected hyper-parameters for each dataset. When selecting the threshold $p$, we start to assign lower values ranging from 0.0 to 0.9 with the step size 0.05. The rationale behind the lowest-to-highest search is that the lowest threshold (even the threshold of 0.0) can still provide efficiency to some extent (as in the analysis of Section 5.3) by transforming the supervision loss into the regularization loss, as similar in previous works (Ye et al., 2021; Modarressi et al., 2022). Additionally, to prevent the router from leaping at the beginning of the training, we initialize the last layer of the routers to favor the non-leap action by setting large biases against the leap action

Table 5: Hyper-parameters of LoT used in each dataset.

| Dataset | Threshold $p$ | Balance term $\lambda$ | Temperature $\tau$ |
|---|---|---|---|
| SST-2 | 0.2 | 2.0 | |
| IMDB | 0.5 | 1.0 | |
| HateXplain | 0.3 | 1.0 | 1.0 |
| AG's news | 0.05 | 4.0 | |
| DBpedia | 0.05 | 4.0 | |
| MRPC | 0.4 | 1.5 | |
| MNLI | 0.5 | 0.9 | 1.0 |
| QNLI | 0.2 | 0.9 | |

## C  Additional Case Study of LoT

In Figure 7, we additionally provide the case study for the two datasets, AG's news and SST-2.

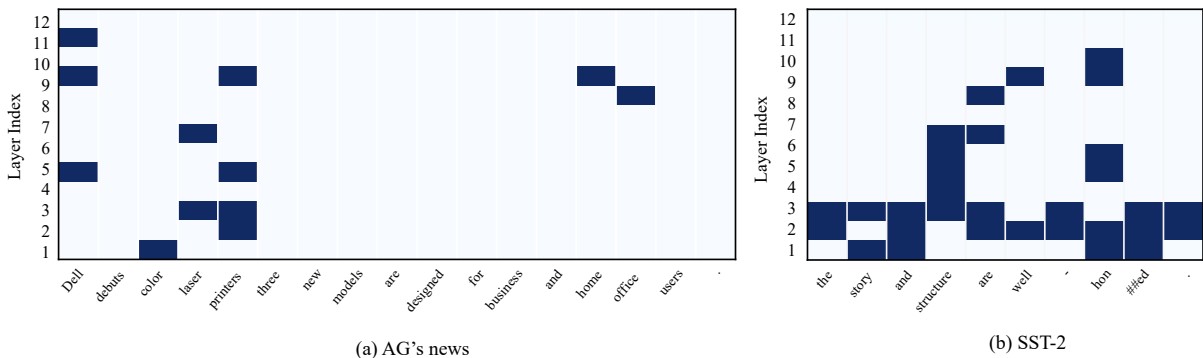

(a) AG's news

(b) SST-2

Figure 7: Illustration of the token routing on two examples. The darker block in each layer indicates the use of corresponding token in the layer while the lighter block denotes the leap action.

## D Computational Overhead

Since LoT requires the dynamic token routers in the transformer, it imposes additional computation cost on our method. This is why we design the router to be a lightweight module, which takes only 2% of the FLOPs from the entire model. Here, we analyze the trade-off between the capacity of the router and total speed-up. Specifically, we set the target performance as fixed and evaluate the total speedup gains with the varying capacity[7] of the router. Figure 8 shows the evaluation results for the trade-off. Notably, increasing the capacity of the router from 0.5% to 2% leads to the improved speedup in both datasets. However, we observe that increasing the computation of the routers to 6% additional FLOPs does not bring speedup gains. This result indicates the router requiring 2% additional FLOPs is enough to achieve reasonable efficiency.

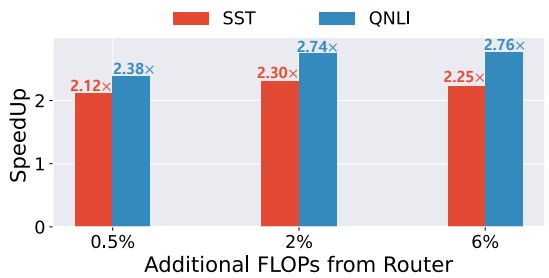

Figure 8: Speedup ratio on the different capacity of dynamic token routers. Note that the router consists of two linear layers.

## E LoT on Different Architectures

To verify the scalability of LoT, we performed the additional experiments on smaller model (i.e., Tiny-

BERT ([Jiao et al., 2020](#))) and larger model (i.e., $\text{BERT}_{\text{large}}$) than the model used in the main paper. Table 6 shows the evaluation results of different scales on SST-2 dataset. The results verify that the proposed method can boost the inference speed on the different scales of PLMs, demonstrating the scalability of LoT.

Table 6: Evaluation results of test accuracy (%) and speedup ratio on the SST-2 dataset.

| Method | Accuracy | SpeedUp |
|---|---|---|
| $\text{BERT}_{\text{large}}$ | 93.5 | 1.00× |
| $\text{BERT}_{\text{large}}$+LoT (ours) | 93.1 | 2.13× |
| TinyBERT | 89.7 | 1.00× |
| TinyBERT+LoT (ours) | 89.5 | 2.22× |

## F Wall-clock Inference Time

To assess the speedup gains on specific computational environments, we measured the inference time on a single NVIDIA V100 GPU. As a result, we observed that the real-time speedup gains (2.2x, Base: 37ms, LoT: 17ms) consist with the gains in FLOPs (2.3x). This observation aligns with the previous finding ([Ye et al., 2021](#)), which suggest that theoretical speedups (i.e., FLOPs) often closely match the actual speedup gains.

---

[7]For the capacity variation, we adjust the dimension of hidden layers of the routers.