# OpenReview forum: "Leap-of-Thought: Accelerating Transformers via Dynamic Token Routing"
_EMNLP/2023/Conference — EMNLP 2023 Main_

### Official Review · Reviewer_K4aR · 2023-08-04

**Paper Topic And Main Contributions:** 1.The paper introduces Leap-of-Though…
**Soundness:** 4

**Excitement:**

3: Ambivalent: It has merits (e.g., it reports state-of-the-art results, the idea is nice), but there are key weaknesses (e.g., it describes incremental work), and it can significantly benefit from another round of revision. However, I won't object to accepting it if my co-reviewers champion it.

**Questions For The Authors:**

1. The Gradient-guided Router is used token-wise instead of token-wise and layer-wise(Equation 7,8 use CAT_i instead of CAT_i^l), is this the reason why the gradient-guidance better than regularization loss enforcement?

**Reasons To Accept:**

1.The gradient-guided router proposed in this paper shows superiority than regularization loss(Figure 5).
2.The result is better than previous token reduction methods.


**Reasons To Reject:**

1.The actual throughputs on specific hardware may need to be examined and compared.
2.The paper only shows the results with BERT base, the performance and speedup with different model size are also considered important considering the model scaling up.



**Reproducibility:**

4: Could mostly reproduce the results, but there may be some variation because of sample variance or minor variations in their interpretation of the protocol or method.

**Reviewer Confidence:**

4: Quite sure. I tried to check the important points carefully. It's unlikely, though conceivable, that I missed something that should affect my ratings.

---

> ### Author Rebuttal · Authors · 2023-08-29
>
> We thank Reviewer K4aR for valuable comments and helpful feedback on our work. We hope our response can address many of the suggestions and concerns.
>
> ### [Actual throughputs on specific hardware]
> Thank you for the suggestion regarding the actual throughputs. We followed the evaluation metrics of the previous SOTA methods (i.e., AdapLeR) that evaluate their methods on FLOPs. To provide additional clarity on the practical efficiency, we measured the inference time on a single NVIDIA V100 GPU. The average inference time on SST-5 as follows.
>
> | Method               | Avg. Inference time |
> |----------------------|:---------------------:|
> | BERT(base)           | 37 ms |
> | BERT(base) + LoT(ours) | 17 ms|
>
> We observed that the real-time speedup gains (2.2x) consist with the gains in FLOPs (2.3x). This observation aligns with previous findings [1,2], which suggest that theoretical speedups (i.e., FLOPs) often closely match the actual speedup gains. We will include these actual speedup results for specific computational environments in Section 4 of the revised paper.
>
> ### [Different model size]
> Thanks for the insightful comment to improve our paper regarding LoT’s applicability. We initially used the BERT base to compare LoT with existing baselines that had evaluated only this model. Following the suggestions, we have performed the additional experiments on the two different models, which are BERT Large and TinyBERT. The experimental results are as follows:
>
> | Method                   | Accuracy | SpeedUp |
> |--------------------------|:----------:|:---------:|
> | BERT(Large)              | 93.5     | 1.00x   |
> | BERT(Large) + LoT (ours) | 93.1     | 2.13x   |
> | TinyBERT                 | 89.7     | 1.00x   |
> | TinyBERT + LoT (ours)    | 89.5     | 2.22x   |
>
> We have observed that applying LoT to two models produces the similar speedup to the BERT base + LoT. We believe these results will help readers better understand the essence of LoT. Therefore, we will incorporate these findings into the updated version of the paper, specifically in Section 5.
>
> ### [Token-wise vs. Token-wise and Layer-wise for Gradient-guided Router]
> Thank you for the attention on the mechanism of the token router. First, we would like to clarify that the router is used both token-wise and layer-wise, similar to positional feed-forward networks in the transformer block. However, as you noticed, the guidance is aggregated from all layers. The rationale behind this decision lies in previous findings on transformer interpretability, which suggest that aggregating gradient information from the entire layers can highlight the important tokens in the input sequence more effectively [3,4]. We believe that this carefully designed guidance produces better performance than the regularization loss, which lacks information about the significance of each token. In the revised version of our paper, we will provide a more detailed background on the rationale behind the token-wise guidance for the router.
>
> ### References
>
> [1] Kim et al., Learned token pruning for transformers, KDD 2022
>
> [2] Ye et al., Tr-bert: Dynamic token reduction for accelerating bert inference, NAACL 2021
>
> [3] Qiang et al., Explaining transformers via attentive class activation tokens, NeurIPS 2022
>
> [4] Barkan et al., Grad-SAM: Explaining Transformers via Gradient Self-Attention Maps, CIKM 2021

---

### Official Review · Reviewer_iScN · 2023-08-09

**Soundness:** 3

**Excitement:**

3: Ambivalent: It has merits (e.g., it reports state-of-the-art results, the idea is nice), but there are key weaknesses (e.g., it describes incremental work), and it can significantly benefit from another round of revision. However, I won't object to accepting it if my co-reviewers champion it.

**Paper Topic And Main Contributions:**

This paper proposes a token reduction strategy for transformer-based models, which aims to improve the computational efficiency and preserve the crucial information for downstream tasks. The main contributions of this paper are: 1. introducing Leap-of-Thought (LoT), a method that enables the dynamic routing of tokens within the transformer layers. LoT learns to decide whether a given token should be processed in the current layer or leaped forward to the next layer, based on a lightweight token router module. 2. The paper proposes a gradient-guided router training, which leverages the gradient information of the token representations to provide supervision for the router. This helps the router to identify and process only the significant tokens for the task at hand. 3. The paper demonstrates the efficacy of LoT through extensive experiments and analysis on various natural language understanding benchmarks, showing that LoT achieves substantial speedup gains without a significant loss in task accuracy.

**Reasons To Accept:**

1. The proposed strategy allows the model to preserve all tokens in subsequent layers while reducing the number of tokens processed within each layer, which can mitigate the risk of losing crucial information and facilitate the exploration of diverse reduction space.
2. The paper proposes a gradient-guided router training, which directly informs each router of which tokens are more influential for the prediction. This can steer the router towards making more informed and efficient decisions about whether to process or leap tokens, as well as provide some degree of interpretability for the model.
3. Authors demonstrate the efficacy of LoT through extensive experiments and analysis on various natural language understanding benchmarks, covering both single input and multiple input tasks. The paper shows that LoT achieves substantial speedup gains over state-of-the-art methods with comparable task accuracy, indicating that LoT can be successfully applied to real-world applications that demand both accuracy and efficiency.
In summary, I recommend this paper should be accepted into Findings.

**Reasons To Reject:**

1. The paper does not provide enough analysis or ablation study on the impact of different components or hyperparameters of LoT, such as the token merging mechanism, the context vector, the threshold p, and the harmony coefficient λ. It would be helpful to understand how these components or hyperparameters affect the performance and behavior of LoT.
2. The paper does not evaluate LoT on natural language generation tasks or LLMs, which could also benefit from improved computational efficiency. It would be valuable to see how LoT performs on these tasks and whether it can preserve the quality and diversity of the generated outputs.

**Reproducibility:**

4: Could mostly reproduce the results, but there may be some variation because of sample variance or minor variations in their interpretation of the protocol or method.

**Reviewer Confidence:**

3: Pretty sure, but there's a chance I missed something. Although I have a good feel for this area in general, I did not carefully check the paper's details, e.g., the math, experimental design, or novelty.

---

> ### Author Rebuttal · Authors · 2023-08-29
>
> We thank Reviewer iScN for valuable comments and helpful feedback on our work. We hope our response can address many of the suggestions and concerns.
>
>
> ### [Additional analysis on the components and hyper-parameters of LoT]
> We appreciate the Reviewer's suggestion for a more in-depth analysis of the components and hyper-parameters of LoT.
> - **Threshold p**: the threshold p represents the cumulative ratio of important tokens over the sequence. Hence the lower the p value forces the model to decide the greater number of tokens to be leaped to the next layer, leading to a higher speedup gain. In contrast, the higher the p value encourages the model to process the greater number of tokens in each transformer block, resulting in a lower speedup gain but higher performance. This hyper-parameter, therefore, controls the trade-off between speed and accuracy. In Section B of the Appendix, we have described the searching process of the hyper-parameter.
> - **Coefficient $\lambda$**: this is the balancing coefficient between the task-specific loss (i.e., cross-entropy) and the token router loss. Its value is empirically determined for each specific task. In our experiments, we initially set a value for a specific threshold and gradually increment the lambda value until we observe a decline in task accuracy.
> - **Merging tokens and context vectors**: The ablation study concerning token merging is already presented in Table 3. As for the context vector, we will add the results (SST-2: 92.7 (2.30x), MRPC: 88.1 (3.11x)) to the same table in the updated paper, indicating a slight performance drop for both tasks. Thank you for bringing attention to our missing ablation study.
> We will include this discussion into Section B of the Appendix, which has described the hyper-parameters in LoT.
>
>
> ### [Evaluations on NLG tasks and LLMs]
> We understand the importance of exploring our method across a broad spectrum of tasks. However, we would like to clarify that the application of LoT to NLG tasks and auto-regressive LLMs, especially for those typically relying on decoder models, is beyond the scope of this work. The nature of generation tasks involves a different set of challenges and would require significant modifications and additional processes to LoT (e.g., cascading attention mechanism, multiple use of token routers in the decoding process). We appreciate your suggestions and plan to investigate this in separate, focused future studies.

---

### Official Review · Reviewer_9vaB · 2023-08-19

**Soundness:** 4

**Excitement:**

3: Ambivalent: It has merits (e.g., it reports state-of-the-art results, the idea is nice), but there are key weaknesses (e.g., it describes incremental work), and it can significantly benefit from another round of revision. However, I won't object to accepting it if my co-reviewers champion it.

**Paper Topic And Main Contributions:**

This paper presents a novel methodology focused on dynamic token routing, a technique devised to identify redundant tokens in the process of sequence understanding inference. Compared to the approaches in prior research, this methodology shows increased flexibility in pinpointing unnecessary tokens, which subsequently enhances efficiency during the inference phase.

This method operates by retaining all input tokens, but under the guidance of a learned router, it adaptively utilizes necessary tokens. This not only allows for a more extensive saving of tokens required for inference but also yields a higher accuracy. An ablation study and a detailed analysis of this approach robustly demonstrate its efficacy.

**Questions For The Authors:**

- I'm curious about the background behind the design choice of having two layers in the Token Router. How are the dimensions of W1 and W2 determined in equation 1? (From what I gather, W2 appears to be (N, 2) to decide whether to leap). Additionally, is there a specific reason for including LayerNorm in between?
- In line 247, what's the purpose of incorporating an additional context vector? The explanation in the main text seems to fall short in clarifying this choice.
- While the benefits of token merging were validated through an ablation study, considering the concept of self-attention, I wonder if this is truly the best approach. When thinking about the self-attention mechanism which measures relative importance between tokens, is merging multiple token representations (even if they've leaped tokens) and including them as a single token in the attention operation truly the optimal way to utilize the information of the leaped tokens? I'd like to hear your thoughts on this.
- As the hidden dimension of the target model increases, I anticipate the training overhead for LoT would also grow more substantial. I'm interested in information regarding the LoT training overhead in relation to varying encoder model sizes.

**Reasons To Accept:**

- The paper is well-structured overall and provides a detailed representation of the proposed methodology, making it easy to follow. The inclusion of descriptions regarding prior research directions that are comparable is commendable, as it facilitated a better comparison of the efficiency of the proposed method.
- By presenting additional analyses, the author provides a deeper understanding of how this method operates in actual natural language understanding scenarios.
- The approach of continuously retaining tokens without discarding them, and making decisions during training on whether to leap over a particular token, is an innovative method compared to related work.

**Reasons To Reject:**

- The diversity of the applied models is limited. It would be beneficial to see how the proposed method performs not only on BERT-base but also on encoders of different sizes (e.g., bigger model: BERT-large, RoBERTa, smaller model: TinyBERT, SkipBERT). Understanding the tendencies across various models (both bigger and smaller) would greatly aid in evaluating and grasping the essence of this methodology.
- While the Speed Up using FLOPS is provided, having comparative data on inference speeds in specific computational environments would offer a clearer picture of the method's practical efficiency.

**Reproducibility:**

4: Could mostly reproduce the results, but there may be some variation because of sample variance or minor variations in their interpretation of the protocol or method.

**Reviewer Confidence:**

4: Quite sure. I tried to check the important points carefully. It's unlikely, though conceivable, that I missed something that should affect my ratings.

**Typos Grammar Style And Presentation Improvements:**

- L274 of of -> of
- L433 Table -> Figure

---

> ### Author Rebuttal · Authors · 2023-08-29
>
> We thank Reviewer 9vaB for valuable comments and pointing out the positive aspects of our work. We hope our response can address many of the suggestions and concerns.
>
> ### [Diversity of the applied models]
> Thank you for the insightful comments to improve our paper, especially concerning LoT's applicability. We initially used the BERT base to compare LoT with existing baselines that had evaluated their methods only on the BERT base model. Following the suggestions, we have performed additional experiments on the two suggested models, which are BERT Large and TinyBERT. The experimental results on SST-5 are as follows:
> | Method                   | Accuracy | SpeedUp |
> |--------------------------|:----------:|:---------:|
> | BERT(Large)              | 93.5     | 1.00x   |
> | BERT(Large) + LoT (ours) | 93.1     | 2.13x   |
> | TinyBERT                 | 89.7     | 1.00x   |
> | TinyBERT + LoT (ours)    | 89.5     | 2.22x   |
>
> We have observed that applying LoT to the two models produces similar speedup to thBERT base + LoT. We believe these results will help readers better understand the essence of LoT. Therefore, we will incorporate these observations into the updated version of the paper, specifically in Section 5.
>
> ### [Real inference speeds in specific computational environments]
> Thank you for the suggestion regarding the practical efficiency. To assess the speedup gains on specific computational environments, we measured the inference time on a single NVIDIA V100 GPU. The average inference time on SST-5 as follows.
>
> | Method               | Avg. Inference time |
> |----------------------|:---------------------:|
> | BERT(base)           | 37 ms |
> | BERT(base) + LoT(ours) | 17 ms|
>
> We observed that the real-time speedup gains (2.2x) consist with the gains in FLOPs (2.3x). This observation aligns with previous findings [1,2], which suggest that theoretical speedups (i.e., FLOPs) often closely match the actual speedup gains. We will include these actual speedup results for specific computational environments in Section 4 of the revised paper.
>
> ### [Design choice about token router]
> Thank you for the detailed attention on the proposed router. The number of layers and the dimension of the token routers was determined through the empirical search. Specifically, we adjust the dimension and the layers by carefully incrementing the capacity of the router, by considering that having more layers and a larger dimension can lead to the increased overheads you mentioned. While the optimal settings can vary based on the difficulty of the tasks, we observed that 2 layers with 256 hidden dimension (i.e., W1=(D, 256), W2=(256, 2) where D is the hidden dimension of the encoder model) generally works well without introducing significant overheads (i.e., 2.1% additional costs for the BERT base and 1.8% for the BERT large). Besides, regarding the inclusion of layer normalization, it was also chosen based on our empirical analysis, where we observed an improved stability and convergence speed during training when including the layer normalization. In the updated paper, we will include the background behind the design choice of the token routers.
>
> ### [Utilizing context vectors in routing decision]
> The context vector was introduced to provide the router with broader context information when deciding on the importance of tokens (we hypothesize that the token importance is determined by its context). Indeed, we empirically observed that making decisions with context vectors results in slightly better accuracy on a few tasks (e.g., removing context vectors in MNLI results in 0.3%p loss in accuracy with similar speedup gains). We will explain it more clearly in the updated version with empirical results.
>
> ### [Design choice of merging tokens]
> We thank the reviewer for detailed attention to the design choice in merging leaped tokens. In the initial version of LoT, the merging mechanism was indeed designed to compute a weighted average of leaped tokens based on their preceding cumulative self-attention scores. However, we found that the performance difference between the self-attention-based average and a simple average was negligible. And that We suspect that such outcome might be due to the fact that, in most cases, the leaped tokens tend to represent less informative portions of the sequence. As a result, their attention scores in relation to each other are relatively uniform, thereby resulting in the simple average during the merging process.
>
> ### [Overhead in Token Router]
> Since the token router uses the hidden states as input, the increased dimension of the encoder model may yet require substantial overhead in the LoT. However, when considering the overhead as a fraction of the overall computations for the encoder, the relative overhead becomes rather smaller, given that the router's overhead increases linearly, whereas the overall computations increase exponentially due to the self-attention and positional feedforward networks. For instance, with a hidden dimension of 256 set for the router (as in our experiment), the computational overhead of the router in the BERT Large model (which has a 1,024 hidden dimension and 24 layers) constitutes only 1.8% of the total computations. This percentage is comparatively lower than the 2.1% overhead in the BERT base model (768 hidden dimension and 12 layers). We appreciate your insights regarding potential limitations tied to the varying sizes of the encoder model. We will include this discussion into the revised version of our paper (Limitation section).
>
> ### [Typos]
> Thank you for pointing out the typos. They will be corrected in the revised paper.
>
> ### References
> [1] Kim et al., Learned token pruning for transformers, KDD 2022
>
> [2] Ye et al., Tr-bert: Dynamic token reduction for accelerating bert inference, NAACL 2021

---

### Meta-Review · Area_Chair_h5E3 · 2023-09-07

**Recommendation:** 4

**Metareview:**

The reviewers agree that the paper is sound and moderately exciting. The routing approach presented is novel and yields good results. The experiments, although limited to a set of few models, are relatively extensive and include all necessary ablations. I recommend acceptance to either the main conference of findings.

---

### Decision · Program_Chairs · 2023-10-07

**Decision:**

Accept-Main

**Comment:**

The reviewers agree that the paper is sound and moderately exciting. The routing approach presented is novel and yields good results. The experiments, although limited to a set of few models, are relatively extensive and include all necessary ablations. I recommend acceptance to either the main conference of findings.